# D3C: Reducing the Price of Anarchy in Multi-Agent Learning

## Abstract

Even in simple multi-agent systems, fixed incentives can lead to outcomes that are poor for the group and each individual agent. We propose a method, D3C, for online adjustment of agent incentives that reduces the loss incurred at a Nash equilibrium. Agents adjust their incentives by learning to mix their incentive with that of other agents, until a compromise is reached in a distributed fashion. We show that D3C improves outcomes for each agent and the group as a whole on several social dilemmas including a traffic network with Braess's paradox, a prisoner's dilemma, and several reinforcement learning domains.

## 1 Introduction

We consider a setting composed of multiple interacting artificially intelligent agents. These agents will be instantiated by humans, corporations, or machines with specific individual incentives. However, it is well known that the interactions between individual agent goals can lead to inefficiencies at the group level, for example, in environments exhibiting social dilemmas (Braess, 1968; Hardin, 1968; Leibo et al., 2017). In order to resolve these inefficiencies, agents must reach a compromise.

Any arbitration mechanism that leverages a central coordinator[1] faces challenges when attempting to scale to large populations. The coordinator's task becomes intractable as it must both query preferences from a larger population and make a decision accounting for the exponential growth of agent interactions. If agents or their designers are permitted to modify their incentives over time, the principal must collect all this information again, exacerbating the computational burden. A central coordinator represents a single point of failure for the system whereas one motivation for multi-agent systems research inspired by nature (e.g., humans, ants, the body, etc.) is robustness to node failures (Edelman and Gally, 2001). Therefore, we focus on decentralized approaches.

A trivial form of decentralized compromise is to require every agent to minimize group loss (maximize welfare). Leaving the optimization problem aside, this removes inefficiency, but similar to a mechanism with a central coordinator, requires communicating all goals between all agents, an expensive step and one with real consequences for existing distributed systems like wireless sensor networks (Kulkarni et al., 2010) where transmitting a signal saps a node's energy budget. There is also the obvious issue that this compromise may not appeal to an individual agent, especially one that is expected to trade its low-loss state for a higher average group loss. One additional, more subtle consequence of optimizing group loss is that it cannot distinguish between behaviors in environments with a group loss that is constant sum, for instance, in zero-sum games. But zero-sum games have rich structure to which we would like agents to respond. Electing a team leader (or voting on a decision) implies one candidate (decision) wins while another loses. Imagine two agents differ on their binary preference with each trying to minimize their probability of losing. A group loss is indifferent; we prefer the agents play the game (and in this, case argue their points).

**Design Criteria**: We seek an approach to compromise in multi-agent systems that applies to the setting just described. The celebrated Myerson-Satterthwaite theorem (Arrow, 1970; Satterthwaite, 1975; Green and Laffont, 1977; Myerson and Satterthwaite, 1983) states that no mechanism exists that simultaneously achieves optimal efficiency (welfare-maximizing behavior), budget-balance (no taxing agents and burning side-payments), appeals to rational individuals (individuals want to opt-in to the mechanism), and is incentive compatible (resulting behavior is a Nash equilibrium). Given

---

[1] For example, the VCG mechanism (Clarke, 1971).

this impossibility result, we aim to design a mechanism that approximates weaker notions of these criteria. In addition, the mechanism should be decentralized, extensible to large populations, and adapt to learning agents with evolving incentives in possibly non-stationary environments.

**Design**: We formulate compromise as agents mixing their incentives with others. In other words, an agent may become incentivized to minimize a mixture of their loss and other agents' losses. We design a decentralized meta-algorithm to search over the space of these possible mixtures.

We model the problem of efficiency using *price of anarchy*. The price of anarchy, $\rho \in [1, \infty)$, is a measure of inefficiency from algorithmic game theory with lower values indicating more efficient games (Nisan et al., 2007). Forcing agents to minimize a group (average) loss with a single local minimum results in a "game" with $\rho = 1$. Note that any optimal group loss solution is also Pareto-efficient. Computing the price of anarchy of a game is intractable in general. Instead, we derive a differentiable upper bound on the price of anarchy that agents can optimize incrementally over time. Differentiability of the bound makes it easy to pair the proposed mechanism with, for example, deep learning agents that optimize via gradient descent (Lerer and Peysakhovich, 2017; OpenAI et al., 2019). Budget balance is achieved exactly by placing constraints on the allowable mixtures of losses. We appeal to individual rationality in three ways. One, we initialize all agents to optimize only their own losses. Two, we include penalties for agents that deviate from this state and mix their losses with others. Three, we show empirically on several domains that opting into the proposed mechanism results in better individual outcomes. We also provide specific, albeit narrow, conditions under which agents may achieve a Nash equilibrium, i.e. the mechanism is incentive compatible, and demonstrate the agents achieving a Nash equilibrium under our proposed mechanism in a traffic network problem.

The approach we propose divides the loss mixture coefficients among the agents to be learned individually; critically, the agents do not need to observe or directly differentiate with respect to the other agent strategies. In this work, we do not tackle the challenge of scaling communication of incentives to very large populations; we leave this to future work. Under our approach, scale can be achieved through randomly sharing incentives according to the learned mixture weights or sparse optimization over the simplex (Pilanci et al., 2012; Kyrillidis et al., 2013; Li et al., 2016).

**Our Contribution:** We propose a differentiable, local estimator of game inefficiency, as measured by price of anarchy. We then present two instantiations of a single decentralized meta-algorithm, one 1st order (gradient-feedback) and one 0th order (bandit-feedback), that reduce this inefficiency. This meta-algorithm is general and can be applied to any group of individual agent learning algorithms.

This paper focuses on how to enable a group of agents to respond to an unknown environment and minimize overall inefficiency. Agents with distinct losses may find their incentives well aligned to the given task, however, they may instead encounter a *social dilemma* (Sec. 3). We also show that our approach leads to interesting behavior in scenarios where agents may need to sacrifice team reward to *save an individual* (Sec. F.4) or need to form parties and *vote* on a new team direction (Sec. 3.4). Ideally, one meta-algorithm would allow a multi-agent system to perform sufficiently well in all these scenarios. The approach we propose, D3C (Sec. 2), is not that meta-algorithm, but it represents a holistic effort to combine critical ingredients that we hope takes a step in the right direction.[2]

## 2 DYNAMICALLY CHANGING THE GAME

In our approach, agents may consider slight re-definitions of their original losses, thereby changing the definition of the original game. Critically, this is done in a way that conserves the original sum of losses (budget-balanced) so that the original group loss can still be measured. In this section, we derive our approach to minimizing the price of anarchy in several steps. First we formulate minimizing the price of anarchy via compromise as an optimization problem. Second we specifically consider compromise as the linear mixing of agent incentives. Next, we define a *local* price of anarchy and derive an upper bound that agents can differentiate. Then, we decompose this bound into a set of differentiable objectives, one for each agent. Finally, we develop a gradient estimator to minimize the agent objectives in settings with bandit feedback (e.g., RL) that enables scalable decentralization.

---

[2]D3C is agnostic to any action or strategy semantics. We are interested in rich environments where high level actions with semantics such as "cooperation" and "defection" are not easily extracted or do not exist.

## 2.1 NOTATION AND TRANSFORMED LOSSES

Let agent $i$'s loss be $f_i(\boldsymbol{x}) : \boldsymbol{x} \in \mathcal{X} \to \mathbb{R}$ where $\boldsymbol{x}$ is the joint strategy of all agents. We denote the joint strategy at iteration $t$ by $\boldsymbol{x}_t$ when considering discrete updates and $\boldsymbol{x}(t)$ when considering continuous time dynamics. Let $f_i^A(\boldsymbol{x})$ denote agent $i$'s transformed loss which mixes losses among agents. Let $\boldsymbol{f}(\boldsymbol{x}) = [f_1(\boldsymbol{x}), \dots, f_n(\boldsymbol{x})]^\top$ and $\boldsymbol{f}^A(\boldsymbol{x}) = [f_1^A(\boldsymbol{x}), \dots, f_n^A(\boldsymbol{x})]^\top$ where $n \in \mathbb{Z}$ denotes the number of agents. In general, we require $f_i^A(\boldsymbol{x}) > 0$ and $\sum_i f_i^A(\boldsymbol{x}) = \sum_i f_i(\boldsymbol{x})$ so that total loss is conserved[3]; note that the agents are simply exploring the space of possible non-negative group loss decompositions. We consider transformations of the form $\boldsymbol{f}^A(\boldsymbol{x}) = A^\top \boldsymbol{f}(\boldsymbol{x})$ where each agent $i$ controls row $i$ of $A$ with each row constrained to the simplex, i.e. $A_i \in \Delta^{n-1}$. $\mathcal{X}^*$ denotes the set of Nash equilibria. $[a; b] = [a^\top, b^\top]^\top$ signifies row stacking of vectors.

## 2.2 PRICE OF ANARCHY

Nisan et al. (2007) define price of anarchy as the worst value of an equilibrium divided by the best value in the game. Here, value means sum of player losses, best means lowest, and Nash is the equilibrium. It is well known that Nash can be arbitrarily bad from both an individual agent and group perspective; Appendix B presents a simple example and demonstrates how opponent shaping is not a balm for these issues (Foerster et al., 2018; Letcher et al., 2018). With the above notation, the price of anarchy, $\rho$, is defined as

$$\rho_{\mathcal{X}}(\boldsymbol{f}^A) \stackrel{\text{def}}{=} \frac{\max_{\mathcal{X}^*} \sum_i f_i^A(\boldsymbol{x}^*)}{\min_{\mathcal{X}} \sum_i f_i^A(\boldsymbol{x})} \geq 1. \tag{1}$$

Note that computing the price of anarchy precisely requires solving for both the optimal welfare and the worst case Nash equilibrium. We explain how we circumvent this issue with a local approximation in §2.4.

## 2.3 COMPROMISE AS AN OPTIMIZATION PROBLEM

Given a game, we want to minimize the price of anarchy by perturbing the original agent losses:

$$\min_{\substack{\boldsymbol{f}' = \psi_A(\boldsymbol{f}) \\ \mathbf{1}^\top \boldsymbol{f}' = \mathbf{1}^\top \boldsymbol{f}}} \rho_{\mathcal{X}}(\boldsymbol{f}') + \nu \mathcal{D}(\boldsymbol{f}, \boldsymbol{f}') \tag{2}$$

where $\boldsymbol{f}$ and $\boldsymbol{f}' = \psi_A(\boldsymbol{f})$ denote the vectors of original and perturbed losses respectively, $\psi_A : \mathbb{R}^n \to \mathbb{R}^n$ is parameterized by weights $A$, $\nu$ is a regularization hyperparameter, and $\mathcal{D}$ penalizes deviation of the perturbed losses from the originals or represents constraints through an indicator function. To ensure minimizing the price of anarchy of the perturbed game improves on the original, we incorporate the constraint that the sum of perturbed losses equals the sum of original losses. We refer to this approach as $\rho$-minimization.

Our agents reconstruct their losses using the losses of all agents as a basis. For simplicity, we consider linear transformations of their loss functions, although the theoretical bounds hereafter are independent of this simplification. We also restrict ourselves to convex combinations so that agents do not learn incentives that are directly adverse to other agents. The problem can now be reformulated. Let $\psi_A(f) = A^\top f$ and $\mathcal{D}(f, f') = \sum_i \mathcal{D}_{KL}(\boldsymbol{e}_i \,||\, A_i)$ where $A \in \mathbb{R}^{n \times n}$ is a right stochastic matrix (rows are non-negative and sum to 1), $\boldsymbol{e}_i \in \mathbb{R}^n$ is a unit vector with a 1 at index $i$, and $\mathcal{D}_{KL}$ denotes the Kullback-Liebler divergence. Note OpenAI Five (OpenAI et al., 2019) also used a linear mixing approach where the "team spirit" mixture parameter ($\tau$) is manually annealed throughout training from 0.3 to 1.0 (i.e., $A_{ii} = 1 - 0.8\tau$, $A_{ij} = 0.2\tau$, $j \neq i$).

The $A$ matrix is interpretable and reveals the structure of "teams" that evolve and develop over training. In experiments we measure *relative reward attention* for each agent $i$ as $\ln((n-1)A_{ii}) - \ln(\sum_{j \neq i} A_{ji})$ to reveal how much agent $i$ attends to their own loss versus the other agents on average (e.g., Figure 4b). This number is 0 when $A_{ij} = \frac{1}{n}$ for all $i, j$. Positive values indicate agent $i$ mostly attends to its own loss. Negative values indicate agent $i$ attends to others' losses more than its own. We also discuss the final $A$ in the election example in §3.4.

---

[3]The price of anarchy assumes positive losses. This is accounted for in §2.5 to allow for losses in $\mathbb{R}$.

### 2.4 A Local Price of Anarchy

The price of anarchy, $\rho \geq 1$, is defined over the joint strategy space of all players. Computing it is intractable for general games. However, many agents learn via gradient-based training, and so only observe the portion of the strategy space explored by their learning trajectory. Hence, we imbue our agents with the ability to locally estimate the price of anarchy along this trajectory.

**Definition 1** (*Local* Price of Anarchy). Define

$$\rho_{\boldsymbol{x}}(\boldsymbol{f}^A, \Delta t) = \frac{\max_{\mathcal{X}_\tau^*} \sum_i f_i^A(\boldsymbol{x}^*)}{\min_{\tau \in [0, \Delta t]} \sum_i f_i^A(\boldsymbol{x} - \tau F(\boldsymbol{x}))} \geq 1 \tag{3}$$

where $F(\boldsymbol{x}) = [\nabla_{x_1} f_1^A(\boldsymbol{x}); \ldots; \nabla_{x_n} f_n^A(\boldsymbol{x})]$, $\Delta t$ is a small step size, $f_i^A$ is assumed positive $\forall i$, and $\mathcal{X}_\tau^*$ denotes the set of equilibria of the game when constrained to the line.

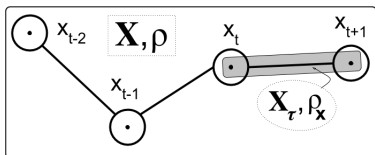

Figure 1: Agents estimate the price of anarchy assuming the joint strategy space, $\mathcal{X}$, of the game is restricted to a local linear region, $\mathcal{X}_\tau$, extending from the currently learned joint strategy, $x_t$, to the next, $x_{t+1}$. $\rho$ and $\rho_x$ denote the global and local price of anarchy.

To obtain bounds, we leverage theoretical results on *smooth games*, summarized as a class of games where "the externality imposed on any one player by the others is bounded" (Roughgarden, 2015). We assume a Lipschitz property on all $f_i^A(x)$ (details in Theorem 1), which allows us to appeal to this class of games. The bound in Eqn (4) is tight for some games. Proofs can be found in appendix D.

**Theorem 1** (Local *Utilitarian* Price of Anarchy). *Assuming each agent's loss is positive and its loss gradient is Lipschitz, there exists a learning rate $\Delta t > 0$ sufficiently small such that, to $\mathcal{O}(\Delta t^2)$, the local **utilitarian** price of anarchy of the game, $\rho_{\boldsymbol{x}}(\boldsymbol{f}^A, \Delta t)$, is upper bounded by*

$$\max_i \{1 + \Delta t \, \texttt{ReLU}\Big(\frac{d}{dt} \log(f_i^A(\boldsymbol{x}(t))) + \frac{||\nabla_{x_i} f_i^A(\boldsymbol{x})||^2}{f_i^A(\boldsymbol{x})\bar{\mu}}\Big)\} \tag{4}$$

*where $i$ indexes each agent, $\bar{\mu}$ is a user defined positive scalar, $\texttt{ReLU}(z) \stackrel{\text{def}}{=} \max(z, 0)$, and Lipschitz implies there exists a $\beta_i$ such that $||\nabla_{x_i} f_i^A(\boldsymbol{x}) - \nabla_{y_i} f_i^A(\boldsymbol{y})|| \leq \beta_i ||\boldsymbol{x} - \boldsymbol{y}|| \; \forall \boldsymbol{x}, \boldsymbol{y}, A$.*[4]

Recall that this work focuses on price of anarchy defined using total loss as the value of the game. This is a *utilitarian* objective. We also derive an upper bound on the local *egalitarian* price of anarchy where value is defined as the max loss over all agents (replace $\sum_i$ with $\max_i$ in Eqn (3); see §D.2).

### 2.5 Decentralized Learning of the Loss Mixture Matrix $A$

Minimizing Eqn (2) w.r.t. $A$ can become intractable if $n$ is large. Moreover, if solving for $A$ at each step is the responsibility of a central authority, the system is vulnerable to this authority failing. A distributed solution is therefore appealing, and the local price of anarchy bound admits a natural decomposition over agents. Equation 2 becomes

$$\min_{A_i \in \Delta^{n-1}} \rho_i + \nu \mathcal{D}_{KL}(\boldsymbol{e}_i \,||\, A_i) \tag{5}$$

where $\rho_i = 1 + \Delta t \, \texttt{ReLU}\Big(\frac{d}{dt} \log(f_i^A(\boldsymbol{x}(t))) + \frac{||\nabla_{x_i} f_i^A(\boldsymbol{x})||^2}{f_i^A(\boldsymbol{x})\bar{\mu}}\Big)$. This objective is differentiable w.r.t. each $A_i$ with gradient $\nabla_{A_i} \rho_i \propto \nabla_{A_i} \texttt{ReLU}\Big(\frac{d}{dt} \log(f_i^A(\boldsymbol{x}(t))) + \frac{||\nabla_{x_i} f_i^A(\boldsymbol{x})||^2}{f_i^A(\boldsymbol{x})\bar{\mu}}\Big)$. The $\log$ appears due to price of anarchy being defined as the worst case Nash total loss *divided* by the minimal total loss. We propose the following modified learning rule for a hypothetical price of anarchy which is

---

[4]Larger $\beta_i$ (less smooth loss) requires smaller $\Delta t$.

defined as a *difference* and accepts negative loss: $A_i \leftarrow A_i - \eta_A \tilde{\nabla}_{A_i} \rho_i$ where $\eta_A$ is a learning rate and

$$\tilde{\nabla}_{A_i} \rho_i = \nabla_{A_i} \text{ReLU}\left(\frac{d}{dt} f_i^A(\boldsymbol{x}) + \epsilon\right). \qquad [\epsilon \text{ is a hyperparameter.}] \qquad (6)$$

The update direction in (6) is proportional to $\nabla_{A_i} \rho_i$ asymptotically for large $f_i^A$; see §D.1.1 for further discussion. Each agent $i$ updates $x_i$ and $A_i$ simultaneously using $\nabla_{x_i} f_i^A(\boldsymbol{x})$ and $\tilde{\nabla}_{A_i} \rho_i$.

**Improve-Stay, Suffer-Shift**—$\nabla_{A_i} \rho_i$ encodes the rule: if the loss is decreasing, maintain the mixing weights, otherwise, change them. This strategy applies Win-Stay, Lose-Shift (WSLS) (Robbins, 1952) to learning (derivatives) rather than outcomes (losses). WSLS was shown to outperform Tit-for-Tat in an iterated prisoner's dilemma (Nowak and Sigmund, 1993; Imhof et al., 2007).

Note that the trival solution of minimizing average group loss coincides with $A_{ij} = \frac{1}{n}$ for all $i, j$. If the agent strategies converge to a social optimum, this is a fixed point in the augmented strategy space $(x, A)$. This can be seen by noting that 1) convergence to an optimum implies $\nabla_{x_i} f_i^A(x) = 0$ and 2) convergence alone implies $\frac{df_i}{dt} = 0$ for all agents so $\nabla A_i = 0$ by Eqn (6) assuming $\epsilon = 0$.

## 2.6 Decentralized Learning & Extending to Reinforcement Learning

The time derivative of each agent's loss, $\frac{d}{dt} f_i^A(\boldsymbol{x}(t))$, in Eqn (6) requires differentiating through potentially all other agent loss functions, which precludes scaling to large populations. In addition, this derivative is not always available as a differentiable function. In order to estimate $\tilde{\nabla}_{A_i} \rho_i$ when only scalar estimates of $\rho_i$ are available as in, e.g., reinforcement learning (RL), each agent perturbs their loss mixture and commits to this perturbation for a random number of training steps. If the loss increases over the trial, the agent updates their mixture in a direction *opposite* the perturbation. Otherwise, no update is performed.

---

**Algorithm 1** D3C Update for RL Agent $i$

Input: $\eta_A, \delta, \nu, \tau_{\min}, \tau_{\max}, A_i^0, \epsilon, l, h, \mathbb{L}$, iterations $T$
$A_i \leftarrow A_i^0$ {Initialize Mixing Weights}
{Draw Initial Random Mixing Trial}
$\tilde{A}_i, \tilde{\boldsymbol{a}}, \tau, t_b, G_b = \texttt{trial}(\delta, \tau_{\min}, \tau_{\max}, A_i, 0, G)$
$G = 0$ {Initialize Mean Return of Trial}
**for** $t = 0 : T$ **do**
  $g = \mathbb{L}_i(\tilde{A}_i \,\forall\, i)$ {Update Policy With Mixed Rewards}
  $\Delta t_b = t - t_b$ {Elapsed Trial Steps}
  $G = (G(\Delta t_b - 1) + g)/\Delta t_b$ {Update Mean Return}
  **if** $\Delta t_b == \tau$ {Trial Complete} **then**
    $\tilde{\rho}_i = \text{ReLU}(\frac{G_b - G}{\tau} + \epsilon)$ {Approximate $\rho$}
    $\nabla_{A_i} = \tilde{\rho}_i \tilde{\boldsymbol{a}} - \nu \boldsymbol{e}_i \oslash A_i$ {Estimate Gradient —(6)}
    $A_i = \text{softmax}_l \lfloor \log(A_i) - \eta_A \nabla_{A_i} \rceil^h$ {Update}
    {Draw New Random Mixing Trial}
    $\tilde{A}_i, \tilde{\boldsymbol{a}}, \tau, t_b, G_b = \texttt{trial}(\delta, \tau_{\min}, \tau_{\max}, A_i, t, G)$
  **end if**
**end for**

---

**Algorithm 2** $\mathbb{L}_i$—example learner

Input: $\tilde{A} = [\tilde{A}_1; \ldots; \tilde{A}_n]$
**while** episode not terminal **do**
  draw action from agent policy
  play action and observe reward $r_i$
  broadcast $r_i$ to all agents
  update policy with $\tilde{r}_i = \sum_j \tilde{A}_{ji} r_j$
**end while**
Output: return over episode $g$

---

**Algorithm 3** $\texttt{trial}$—helper function

Input: $\delta, \tau_{\min}, \tau_{\max}, A_i, t, G$
{Sample Perturbation Direction}
$\tilde{\boldsymbol{a}}_i \sim U_{sp}(n)$
{Perturb Mixture}
$\tilde{A}_i = \text{softmax}(\log(A_i) + \delta \tilde{\boldsymbol{a}}_i)$
{Draw Random Trial Length}
$\tau \sim \text{Uniform}\{\tau_{\min}, \tau_{\max}\}$
Output: $\tilde{A}_i, \tilde{\boldsymbol{a}}, \tau, t, G$

---

This is formally accomplished with approximate one-shot gradient estimates (Shalev-Shwartz et al., 2012). A one-shot gradient estimate of $\rho_i(A_i)$ is performed by first evaluating $\rho_i(\log(A_i) + \delta \tilde{\boldsymbol{a}}_i)$ where $\delta$ is a scalar and $\tilde{\boldsymbol{a}}_i \sim U_{sp}(n)$ is drawn uniformly from the unit sphere in $\mathbb{R}^n$. Then, an unbiased gradient is given by $\frac{n}{\delta} \rho_i(\log(A_i) + \delta \tilde{\boldsymbol{a}}_i) \tilde{\boldsymbol{a}}_i$ where $A_i \in \Delta^{n-1}$. In practice, we cannot evaluate in one shot the $\frac{d}{dt} f_i^A(\boldsymbol{x}(t))$ term that appears in the definition of $\rho_i$. Instead, Algorithm 1 uses finite differences and we assume the evaluation remains accurate enough across training steps.

Algorithm 1 requires arguments: $\eta_A$ is a global learning rate for each $A_i$, $\delta$ is a perturbation scalar for the one-shot gradient estimate, $\tau_{\min}$ and $\tau_{\max}$ specify the lower and upper bounds for the duration of the mixing trial for estimating a finite difference of $\frac{d}{dt} f_i^A(\boldsymbol{x}(t))$, $l$ and $h$ specify lower and upper

bounds for clipping $A$ in logit space $({}_l\lfloor\cdot\rceil^h)$, and $\mathbb{L}_i$ is a learning algorithm that takes $A$ as input (in order to mix rewards) and outputs *discounted return*. $\oslash$ indicates elementwise division.

## 2.7 ASSESSMENT

We assess Algorithm 1 with respect to our original design criteria. As described, agents perform gradient descent on a decentralized and local upper bound on the price of anarchy. Recall that a minimal global price of anarchy ($\rho = 1$) implies that even the worst case Nash equilibrium of the game is socially optimal; similarly, Algorithm 1 searches for a locally socially optimal equilibrium. By design, $A_i \in \Delta^{n-1}$ ensures the approach is budget-balancing. We justify the agents learning weight vectors $A_i$ by initializing them to attend primarily to their own losses as in the original game. If they can minimize their original loss, then they never shift attention according to Eqn (6) because $\frac{df_i}{dt} \leq 0$ for all $t$. They only shift $A_i$ if their loss increases. We also include a KL term to encourage the weights to return to their initial values. In addition, in our experiments with symmetric games, learning $A$ helps the agents' outcomes in the long run. We also consider experiments in Appendix E.2.2 where only a subset of agents opt into the mechanism. If each agent's original loss is convex with diagonally dominant Hessian and the strategy space is unconstrained, the unique, globally stable fixed point of the game defined with mixed losses is a Nash (see Appendix H.4). Exact gradients $\nabla_{A_i}\rho_i$ require each agent differentiates through all other agents losses precluding a fully decentralized and scalable algorithm. We circumvent this issue with noisy oneshot gradients. All that is needed in terms of centralization is to share the mixed scalar rewards; this is cheap compared to sharing $x_i \in \mathbb{R}^d$. As mentioned in the introduction, the cost of communicating rewards can be mitigated by learning $A_i$ via sparse optimization or sampling but is outside the scope of this paper.

## 3 EXPERIMENTS

Here, we show that agents minimizing local estimates of price of anarchy achieve lower loss on average than selfish, rational agents in three domains. Due to space, we leave two other domains to the appendix. In the first domain, a traffic network (4 players), players optimize using exact gradients (see Eqn (6)). Then in two RL domains, Coins and Cleanup, players optimize with approximate gradients as handled by Algorithm 1. Agents train with deep networks and A2C (Espeholt et al., 2018). We refer to both algorithms as D3C (decentralized, differentiable, dynamic compromise).

For D3C, we initialize $A_{ii} = 0.99$ and $A_{ij} = \frac{0.01}{n-1}$, $j \neq i$. We initialize away from a onehot because we use entropic mirror descent (Beck and Teboulle, 2003) to update $A_i$, and this method requires iterates to be initialized to the interior of the simplex. In the RL domains, updates to $A_i$ are clipped in logit-space to be within $l = -5$ and $h = 5$ (see Algorithm 1). We set the $\mathcal{D}_{KL}$ coefficient to 0 except for in Coins, where $\nu = 10^{-5}$. Additional hyperparameters are specified in §G. In experiments, reported price of anarchy refers to the ratio of the sum of losses of the strategy learning converged to over that of the strategy learned by fully cooperative agents ($A_{ij} = \frac{1}{n}$).

## 3.1 TRAFFIC NETWORKS AND BRAESS'S PARADOX

In 2009, New York city's mayor closed Broadway near Times Square to alleviate traffic congestion (Neuman and Barbaro, 2009). This counter-intuitive phenomenon, where restricting commuter choices improves outcomes, is called Braess's paradox (Wardrop, 1952; Beckmann et al., 1956; Braess, 1968), and has been observed in real traffic networks (Youn et al., 2008; Steinberg and Zangwill, 1983). Braess's paradox is also found in physics (Youn et al., 2008), decentralized energy grids (Witthaut and Timme, 2012), and can cause extinction cascades in ecosystems (Sahasrabudhe and Motter, 2011). Knowing when a network may exhibit this paradox is difficult, which means knowing when network dynamics may result in poor outcomes is difficult.

Figure 2a presents a theoretical traffic network. Without edge AB, drivers commute according to the Nash equilibrium, either learned by gradient descent or D3C. Figure 3a shows the price of anarchy approaching 1 for both algorithms. If edge AB is added, the network now exhibits Braess's paradox. Figure 3b shows that while gradient descent converges to Nash ($\rho = \frac{80}{65}$), D3C achieves a price of anarchy near 1. Figure 2b shows that when faced with a randomly drawn network, D3C agents achieve shorter commutes on average than agents without the ability to compromise.

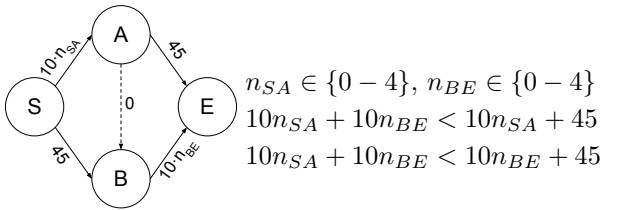 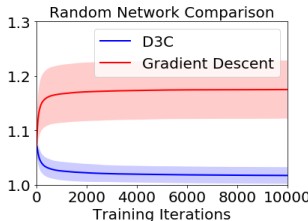

(a) Traffic Network       (b) Random Network Results

Figure 2: (a) Four drivers aim to minimize commute time from S to E. Commute time on each edge depends on the number of commuters, $n_{ij}$. Without edge AB, drivers distribute evenly across SAE and SBE for a 65 min commute. After edge AB is added, switching to the shortcut, SABE, always decreases commute time given the other drivers maintain their routes, however, all drivers are incentivized to take the shortcut resulting in an 80 min commute. (b) The price of anarchy throughout training for 1000 randomly generated three road networks exhibiting Braess's paradox. Gradient descent and D3C are compared with shaded regions representing $\pm 1$ standard deviation.

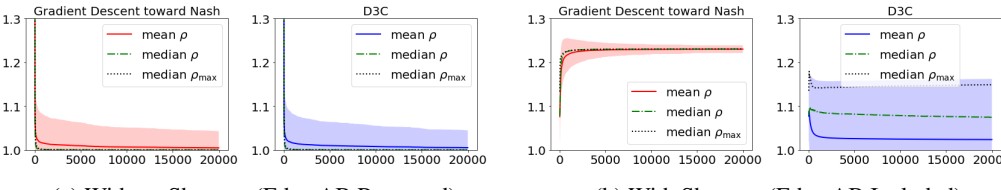

(a) Without Shortcut (Edge AB Removed)      (b) With Shortcut (Edge AB Included)

Figure 3: **Traffic Network** (a) Without edge AB, agents are initialized with random strategies and train with either gradient descent (left) or D3C (right)—similar performance expected. Statistics of 1000 runs are plotted over training. Median $\rho_{\max}$ tracks the median over trials longest-commute among the four drivers. The shaded region captures $\pm 1$ standard deviation around the mean. (b) After edge AB is added, agents are initialized with random strategies and trained with either gradient descent (left) or D3C (right). Statistics of 1000 runs are plotted over training. Median $\rho_{\max}$ tracks the median over trials of the longest-commute among the four drivers. The shaded region captures $\pm 1$ standard deviation around the mean.

### 3.2 COIN DILEMMA

In the Coins game (Eccles et al., 2019a; Lerer and Peysakhovich, 2017), two agents move on a fully-observed $5 \times 5$ gridworld, on which coins of two types corresponding to each agent randomly spawn at each time step with probability $0.005$. When an agent moves into a square with a coin of either type, they get a reward of $1$. When an agent picks up a coin of the other player's type, the other agent gets a reward of $-2$. The episode lasts $500$ steps. Total reward is maximized when each agent picks up only coins of their own type, but players are tempted to pick up all coins.

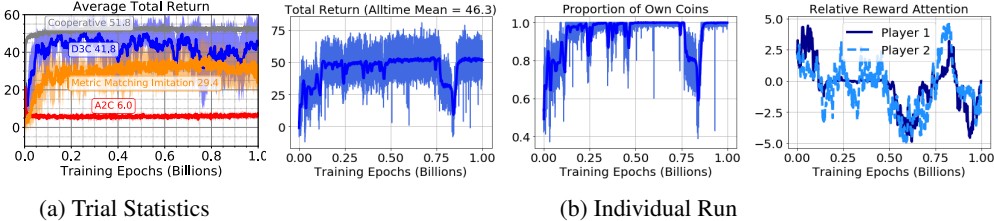

(a) Trial Statistics       (b) Individual Run

Figure 4: **Coin Dilemma** (a) Mean total return over ten training runs for agents. Mean return over all epochs is next to each method name in the legend. D3C hyperparameters were selected using five independent validation runs. Cooperative agents trained to maximize total return represent the best possible baseline. Shaded region captures $\pm 1$ standard deviation around the mean. (b) One training run ($A_{ii}^0 = 0.9$): sum of agent returns (left); % of coins picked up that were the agent's type (middle); relative reward attention measured as $\ln((n-1)A_{ii}) - \ln(\sum_{j \neq i} A_{ji})$ (right).

D3C agents approach optimal cooperative returns (see Figure 4a). We compare against Metric Matching Imitation (Eccles et al., 2019b), which was previously tested on Coins and designed to exhibit reciprocal behavior towards co-players.

Figure 4b shows D3C agents learning to cooperate, then temporarily defecting before rediscovering cooperation. Note that the relative reward attention of both players spikes towards selfish during this small defection window; agents collect more of their opponent's coins during this time. Oscillating between cooperation and defection occurred across various hyperparameter settings. Relative reward attention trajectories between agents appear to be reciprocal, i.e., move in relative synchrony (see §H.2 for analysis).

## 3.3 CLEANUP

We provide additional results on Cleanup, a five-player gridworld game (Hughes et al., 2018). Agents are rewarded for eating apples, but must keep a river clean to ensure the apples receive sufficient nutrients. The option to be a freeloader and only eat apples presents a social dilemma. D3C is able to increase both welfare and individual reward over A2C (no loss sharing). We also observe that direct welfare maximization (Cooperation) always results in three agents collecting rewards from apples while two agents sacrifice themselves and clean the river. In contrast, D3C avoids this stark division of labor. Agents take turns on each task and all achieve some positive cumulative return over training.

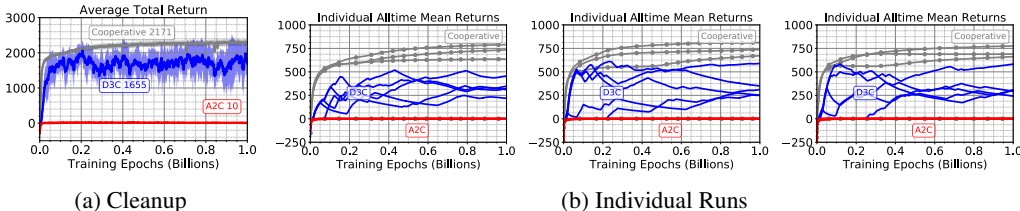

(a) Cleanup            (b) Individual Runs

Figure 5: (a) Mean total return over ten training runs for agents. D3C hyperparameters were selected using five independent validation runs. Cooperative agents trained to maximize total return represent the best possible baseline. Shaded region captures $\pm 1$ standard deviation around the mean. (b) Three randomly selected runs. Each curve shows the mean return up to the current epoch for 1 of 5 agents.

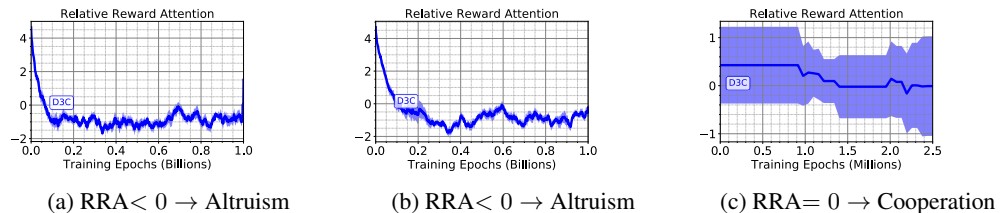

(a) RRA$< 0 \rightarrow$ Altruism     (b) RRA$< 0 \rightarrow$ Altruism     (c) RRA$= 0 \rightarrow$ Cooperation

Figure 6: Average relative reward attention (RRA) over 10 runs for five agents in Cleanup (a) and Harvest Patch (b) and two agents in Mini-Cleanup (c).

## 3.4 A ZERO-SUM ELECTION

Consider a hierarchical election in which two parties compete in a zero-sum game—for example, only one candidate becomes president. If, at the primary stage, candidates within one party engage in negative advertising, they hurt their chances of winning the presidential election because these ads are now out in the open. This presents a prisoner's dilemma within each party. The goal then is for each party to solve their respective prisoner's dilemma and come together as one team, but certainly not maximize welfare—the zero-sum game between the two parties should be retained. A simple simulation with two parties consisting of two candidates each initially participating in negative advertising converges to the desired result after running D3C.

The final $4 \times 4$ loss mixing matrix, $A$, after training 1000 steps is an approximate block matrix with 0.46 on the $2 \times 2$ block diagonal and 0.04 elsewhere. We make a *duck-typing* argument that when

multiple agents are optimizing the same loss, they are functioning as multiple components of a single agent because mathematically, there is no difference between this multi-agent system and a single agent optimization problem. This matrix then indicates that two approximate teams have formed: the first two agents captured by the upper left block and vice versa. Furthermore, the final eigenvalues of the game Jacobian are $(1.84 \pm 0.21i) \times 2$; perfect team formation gives $(2 \pm 0.25i) \times 2$. The existence of imaginary eigenvalues indicates that the zero-sum component of the game is retained. In contrast, minimizing total loss gives $0$ imaginary part because Hessians ($\mathtt{Jac}(\nabla)$) are symmetric.

## 4 RELATED WORK

Our work is most similar to (Hostallero et al., 2020). This work also provides a decentralized approach that transforms the game by modifying rewards, however, it does not guarantee "budget-balance" nor does it derive its proposed algorithm from any principle (e.g., price of anarchy); the proposed algorithm is a heuristic supported by experiments. In other work, Lupu and Precup (2020) explore gifting rewards to agents as well, but it does so by simply expanding the action space of agents to include a gifting action. It is also not budget balanced.

### 4.1 LEARNING LOSS FUNCTIONS

Choosing the right loss function for a given task is a historically unsolved problem. Even in single-agent settings, the designated reward function can either often be suboptimal for learning (Sorg et al., 2010) or result in "reward hacking" (Amodei et al., 2016). In the multiagent setting, OpenAI Five trains on an objective that mixes single agent and group agent rewards (OpenAI et al., 2019). The "team spirit" mixture parameter ($\tau$) is manually annealed throughout training from 0.3 to 1.0 (i.e., $A_{ii} = 1 - 0.8\tau, A_{ij} = 0.2\tau, j \neq i$). Liu et al. (2019) find a team of soccer agents is better trained with agent-centric shaping rewards evolved via population based training, a technique that also led to human level performance in Capture the Flag (Jaderberg et al., 2019). Aguera y Arcas (2020) trains populations of simulated bacteria to maximize randomly drawn reward functions and discovers that a significant portion of the surviving populations are actually ones rewarded for dying. In contrast to the work just described, we provide a decentralized approach, devoid of a central authority, that automates the design of incentives for a multi-agent system.

### 4.2 SOCIAL PSYCHOLOGY, NEUROSCIENCE, AND EVOLUTIONARY BIOLOGY

Loss transformation is also found in human behavior. Within social psychology, *interdependence theory* (Kelley and Thibaut, 1978) holds that humans make decisions based on a combination of self interest and social preferences. In game theoretic terms, humans deviate from rational play because they consider a transformed game rather than the original. Although rational play in the transformed game may result in lower payoff in a single round of play, groups with diverse transformations are oftentimes able to avoid poor Nash equilibria. McKee et al. (2020) mirrored this result empirically with RL agents. Neuroscience research also supports this interpretation, showing that neural processing responds to others' losses, even if one's own outcomes are not affected (Fukushima and Hiraki, 2009; Kang et al., 2010). The most fundamental account within evolutionary biology predicts that nature selects for individuals who care only for their own fitness. Absent other mechanisms, local selection for selfishness can drive a population to extinction (Grande-Pérez et al., 2005). The emergence of other-regarding preferences seems particularly important for humans. Empathy results in altruistic choices, raising the fitness of their group as a whole (Alexander and Bargia, 1978).

## 5 CONCLUSION

Directly maximizing welfare can solve many social dilemmas, but it fails to draw out the rich behavior we would expect from agents in other interesting scenarios. We formulate learning incentives as a price of anarchy minimization problem and propose a decentralized, gradient-based approach, namely D3C, that incrementally adapts agent incentives to the environment at hand. We demonstrate its effectiveness on achieving near-optimal agent outcomes in socially adversarial environments. Importantly, it also generates reasonable responses where welfare maximization is indifferent.

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
