# OpenReview forum: "D3C: Reducing the Price of Anarchy in Multi-Agent Learning"
_ICLR.cc/2021/Conference — Reject_

### Official Review · AnonReviewer1 · 2020-10-20
**I'm confused about too many things with this paper**

**Rating:** 3
**Confidence:** 3

**Review:**

The authors propose to change agents' incentives in order to lead to a better Nash equilibrium outcome.
They propose a practical decentralized approach to computing the "optimal" change of incentives.
I'm confused by many aspects of this paper:
1. How would you implement the change of incentives in a any of the game theoretic domains? I.e. in what context would agents who lose money agree to participate without a centralized authority ordering them to?
2. Why do you consider changes of incentives with bounded KL-divergence? What does that have to do with your motivating concern around communication (3rd paragraph of intro)?
3. Why do you optimize the price of anarchy instead of social welfare at worst Nash equilibrium? These should be the same when you don't change the total loss, but then I don't understand the denominator of (3).
4. What is $t$ in (4)? How does f^A_i depend on t?
5. Shouldn't \beta_i appear in (4)? Or does (4) hold for all \beta_i (e.g. approaching infinity)?
6. What should I conclude from your experiments? Of course if you completely mix the agents' utilities the Nash equilibrium would be locally optimal. The Nash equilibrium utilities at the modified games seem better, but it's not clear how much the games were modified.

Misc:
(A) Computing the Price of Anarchy in general is NP-complete rather than PPAD-complete because PPAD corresponds to finding *any* Nash equilibrium rather than the worst one.
(B) "set of the equilibria of the game restricted to the line" - I don't understand what this sentence means.
(C) Why don't you define utilitarian/egalitarian before Theorem 1?
(D) Typos: "one-hot" "the the"

---

> ### Author Response · Authors · 2020-11-16
> **Response to Reviewer 1**
>
> Thank you for reviewing our paper. We will do our best to clear up the points of confusion you note above. Please take these into consideration when determining how to adjust your score. We will respond bullet by bullet.
>
> 1.
> a) Game Theoretic Domains -- Do you mean, for example, $2 \times 2$ games like chicken or matching pennies? We explore a game of this form in Section F.4. These classic normal form games are also known as linear bimatrix games and can be written as $\max_{x \in \Delta} x^T A y = u_1(x,y)$ and $\max_{y \in \Delta} x^T B y = u_2(x,y)$ where $\Delta$ denotes the probability simplex. That is to say they are no different in form than the differentiable prisoner's dilemma game we explore in Section E.2. Utility functions just need to be negated to be reinterpreted as losses.
> b) Reviewer 4 made a similar comment. Please take a second look at the introduction where we discuss how "We appeal to individual rationality in three ways." Also look at Section 2.7 starting with "We justify the agents learning weight vectors..." which explains how the proposed meta-algorithm addresses these concerns. We also point out in Appendix E.2.2 that D3C agents are robust to “mavericks” that choose not to mix losses in the prisoner’s dilemma domain. Please let us know any specific questions you have after reviewing these.
>
> 2.
> The $KL$ term does not have to do with the communication concern, but rather with the rationality concern you raise above. An agent that learns to be altruistic (i.e., one that gives its reward to another agent) might be easily taken advantage of. Adding a $KL$ penalty encourages the agent to selfishly attend to its own loss so that it avoids altruistic behaviors if unnecessary.
>
> 3.
> Your understanding of Equation 1 is correct. The denominator is a constant given total loss is conserved (i.e., budget balanced). If we could just minimize the numerator, we could directly minimize global PoA. However, minimizing the numerator requires computing the Nash equilibrium which is intractable. Instead, we introduce local PoA where we constrain the joint strategy space to a line segment. For example, consider playing rock paper scissors and then being told you must play rock, paper, and scissors such that $a \cdot p_{rock} + b \cdot p_{paper} + c \cdot p_{scissors} = d$. We have now constrained your strategy space. By constraining the strategy space and leveraging theory from smooth games, we're able to construct an upper bound on this local PoA that circumvents directly computing a Nash. This is not necessarily the only way to simplify the game such that local PoA can be efficiently approximated. Alternative approaches are work for future research.
>
> 4.
> Let $x=x(t)$ be a function that returns the joint strategy $x$ along any point in its trajectory. Gradient descent dynamics state that $\dot{x} = - \frac{df_i^A(x)}{dt}$. These are the continuous time dynamics of $x$. $f^A_i(x)$ then depends on $t$ through $x(t)$. We will add text to make this dependence more clear.
>
> 5.
> Yes, there is some dependence of Equation 4 on $\beta$ which is resolved by using a smaller $\Delta t$ (i.e., policy learning rate). The larger the $\beta$, the smaller the $\Delta t$ that is required for the bound to hold. This is a technical detail that is identified more clearly in the derivation in the appendix. For example, check out Equation 63 in Lemma 7 of Section D and note that $\xi$ contains a $\beta$ term. If $\beta_i$ is very large, then $\Delta t$ has to be very small to ensure $a_i$ is greater than $0$.
>
> 6.
> Our experiments are meant to support the “Our Contribution” statement made in the Intro. You should conclude that the agents are able to minimize the price of anarchy (thereby e.g., resolving social dilemmas) in a decentralized fashion using only access to the other agents’ scalar reward signals. Section 3.4 is a counterargument to "completely mix the agents' utilities". Uniform mixing is not the answer to all multiagent problems. We present an algorithm that automatically learns how to mix depending on the multiagent problem domain. This is also summarized in our Conclusion.
>
> Misc:
>
> (A) Thank you. That's a great point. We'll make the change.
>
> (B) Redefine the game with player strategies restricted to the line segment $\mathcal{X}_{\tau}$. All other components of the game (utilities, players, etc.) stay the same. We give a more detailed response above in bullet 3.
>
> (C) The utilitarian / egalitarian distinction is not central to the paper. This is more of a footnote to point the reader to Appendix D.2 if they are interested.

---

### Official Review · AnonReviewer3 · 2020-10-27
**Nice idea, but not explored deeply enough**

**Rating:** 6
**Confidence:** 2

**Review:**

The paper details a method by which it attempts a method to change the utility function for agents so they incorporate the utilities of other agents, resulting in more cooperating agents, so that the price of anarchy is minimized. It examines such trained agents in several problem settings, seeing them performing quite well.

The attempt to change the utility is quite interesting, but, sadly, it is not really explored enough. That is, while agents improve, no attempt has been made to examine the new "mixing" matrix, and see what changed to make the agents collaborate better (or not). It is quite a pity, as in this case, unlike many other learning cases, the result is quite interpretable for us.

As more a game-theorist than a ML expert, I did struggle a bit with the technical bits in section 2. The definition of "local price of anarchy" seems sometime overlapping, but not really the same as the definition of that of Ben-Zwi and Ronen from 2011 ("Local and global price of anarchy of graphical games", TCS 412, 1196-1207). I had some difficulty following algorithm 1, as \Delta t_b seems always negative, and it was not clear to me what G is (and in particular, for negative values).

In the experimental section, as noted above, an interpretation of the A matrix would have helped. At least in some of the examples, it seems a matrix A with each value being 1/n seems to satisfy all that's needed (e.g. the Braess paradox). I could not follow the setting of the "zero-sum election", and what is the described game. Moreover, the introduction promises an incentive-compatible setting. What is intended to be an IC mechanism?

UPDATE POST-REBUTTAL
————————————
Many of my questions have been answered, though I do think reviewers should explicitly note Ben-Zwi and Ronen's paper, and change their use of "incentive compatible". I think a deeper and more systematic analysis of the A matrix is also warranted, but I do now feel the paper has better scientific merit.

---

> ### Author Response · Authors · 2020-11-16
> **Response to Reviewer 3**
>
> Thank you for reviewing our paper. From our experience, the bulk of recent work in social dilemmas in deep (neural nets) multi-agent RL fails to explain its contributions in light of longstanding issues already raised by game theory, and so we made a concerted effort here to tie our work to game theoretic concepts and terminology (e.g., price of anarchy, Myerson-Satterthwaite theorem, etc.). These fields are evolving somewhat independently, so tying them together requires finesse. As someone with a game theory background, we very much appreciate your viewpoint here.
>   Also, we believe that a few of your major concerns, for example, regarding the lack of analysis of the “mixing matrix” and the overlap of local PoA with other work, can be easily cleared up. Please see our responses below. We hope that in light of these clarifications (specifically regarding the analysis of the mixing matrix), you will consider reassessing your score.
>
> “Mixing” Matrix -- This is a simple misunderstanding. We completely agree with you. The mixing matrix is very interesting to analyze, and we do in fact examine the "mixing" matrix in 4 cases, although maybe it was not obvious to all reviewers.
>   Figure 4b "Relative Reward Attention" plots $\log(A_{ii} / A_{ji})$ for each player in the Coin dilemma. Note that for a $2 \times 2$ matrix, each player's mixing weight can be uniquely represented by 1 number because $A_{ij} = 1 - A_{ii}$ by the simplex constraint. The relative reward attention plot shows the agents start off selfish (positive values) and then become cooperative (zero), then altruistic (negative), and then oscillate.
>   This same type of plot is included for the Prisoner’s Dilemma experiment in the appendix. Figure 10 "Relative Loss Attention" (RLA) indeed shows the agents converge to the cooperative solution of $\frac{1}{n}$ as you expected! We’ll make it more clear that the RLA asymptote of $0$ for $n=2$ agents corresponds to $A = \frac{1}{n}$ here. The fact that the trivial solution of $A = \frac{1}{n}$ always exists is discussed in the Intro (paragraph 3), the last paragraph of Section 2.5, and the second paragraph of Section 3.
>   “Relative Loss Attention” is also plotted for the “Trust-Your-Brother” domain in Figure 12.
>   We did not include this same plot for Mini-Cleanup (Figure 14) or Harvest Patch (Figure 15), but we have these plots ready for the revision. The oscillation of relative reward attention is what gives rise to the alternating of roles in the cleanup domain (Figure 5) and allows agents to achieve similar levels of reward. This is in contrast to a stark division of labor seen in other work (see F.2 which discusses Yang et al. '20).
>   We also explicitly discuss the resulting "mixing" matrix in Section 3.4 for the election example. That section is specifically included to emphasize the value of analyzing the "mixing" matrix. We can include a diagram that better illustrates the hierarchical game being played here. Abstractly, it looks like
>
>                                        Party A vs Party B (Zero-Sum)
>
>                            //                                                \\
>
>           Party A Primary (Social Dilemma)                 Party B Primary (Social Dilemma)
>
>          //                             \\                  //                         \\
>       Candidate 1                Candidate 2          Candidate 3                Candidate 4
>
> and mathematically it is created by summing the utilities defining a zero-sum game over all 4 agents with the utilities defining two social dilemmas (specifically the prisoner’s dilemma utilities defined in Section E.2) defined over the two separate parties A and B.
>
> Local PoA [Ben-Zwi and Ronen ‘11] -- We agree. The definition of local in the paper you reference appears to be overlapping with ours. Their notion of local considers a subset of the strategy space that only contains a subset of the players (specifically the neighboring nodes in the graphical game). This would equate to considering only subspaces (specifically axis aligned projections) of the joint strategy space $\mathcal{X}$ in our formulation whereas we do not make this restriction. On the other hand, we do not consider the entire strategy space either. Local, for us, generally considers all players in the game, but restricts the pure strategies (actions) allowed by each player. So, in summary, these two definitions look at two ways of defining "local".
>   For concreteness consider a two player, two action game. Each player $i$’s mixed strategy can be represented by a single probability scalar $p_i$. The joint strategy space can be visualized as a unit square in the positive quadrant. The paper you reference considers a local game to be either constrained to the x-axis or y-axis. Our definition considers any small line segment within the unit square. We can include such a diagram and explanation in the appendix.

---

> > ### Author Response · Authors · 2020-11-16
> > **Response to Reviewer 3 (continued)**
> >
> > Algorithm 1 --  We break this response into 3 parts.
> > a) $\Delta t_b$ is, in fact, always positive. $t_b$ is returned by the function $trial$ at the end of the for-loop which sets $t_b$ to the current time step $t$. The quantity $t-t_b$ evaluated at the start of the subsequent loop then represents the elapsed time, a positive value.
> > b) $G$ tracks the mean of $g$. $g$ is the return (sum of discounted rewards). This scalar can be positive or negative and depends on the environment. We use $G$ as it is sometimes the variable used for returns in the RL literature.
> > c) To give you a high level understanding of the algorithm, it may be helpful to read the orange comments in the algorithm block. Each agent draws a mixing vector (from trial) and tries optimizing their policy with respect to their reward mixed according to all the agents' current mixing weights. Think of this as "testing out an alliance". If their reward signal (i.e., return or $G$ or negative loss or utility) decreases on average over the span of the trial, the agent adjusts their mixing weight in the opposite direction (because that alliance made them worse off). This loop is repeated over training.
> >
> > Incentive Compatibility -- The meta-algorithm that adjusts the mixing weights is the mechanism just as Vickrey-Clarke-Groves (VCG) is a mechanism, however, in contrast to VCG, our meta-algorithm is distributed over agents and can be viewed as the agents negotiating their own compromise. If you mean that we do not explore settings where agents are capable of explicitly "lying", that is true. However, an agent may mislead another by adjusting its mixing weights and use that to its advantage. With this in mind, agents converge to a stable equilibrium in the Braess' paradox example with $A=\frac{1}{n}$ for which any strategy that attempts to mislead (lie) by deviating is immediately punished. Therefore, truthfulness is ultimately incentivized in that example.

---

> > > ### Comment · AnonReviewer3 · 2020-11-21
> > > **Zero Sum and IC**
> > >
> > > Thank you — you answered a lot of my questions. However, I'd like some clarification on two points:
> > >
> > > 1. I could not follow the description of the zero-sum election. What are the player's strategies?
> > >
> > > 2. You claim, in the introduction, "We also provide specific, albeit narrow, conditions under which agents may achieve a Nash equilibrium, i.e. the mechanism is incentive compatible". First, a mechanism with a NE is not necessarily incentive compatible. Second, where are these specific conditions? The words "incentive compatible" do not occur in the paper beyond the introduction.

---

> > > > ### Author Response · Authors · 2020-11-23
> > > > **Re "Zero Sum and IC"**
> > > >
> > > > 1.
> > > > Each player’s strategy, $x_i$, is a 1-d scalar variable constrained to [0, 1], so it can be interpreted as a probability if that is most comfortable, but in truth it is just a scalar belonging to a convex set. We did not create them with any semantics in mind. We construct the losses by adding the losses from the prisoner’s dilemma game (see Figure 8 on p. 23 in the supplementary) and a canonical zero-sum game ($\min_{x_1, x_2} \max_{x_3, x_4} [x_1, x_2]^\top [x_3, x_4]$). The losses are defined for each "candidate" as follows:
> > > > - $f_1(x) = x_1^2 + (x_2 - 1)^2 + x_1 x_3 + x_2 x_4$
> > > > - $f_2(x) = x_2^2 + (x_1 - 1)^2 + x_1 x_3 + x_2 x_4$
> > > > - $f_3(x) = x_3^2 + (x_4 - 1)^2 - x_1 x_3 - x_2 x_4$.
> > > > - $f_4(x) = x_4^2 + (x_3 - 1)^2 - x_1 x_3 - x_2 x_4$.
> > > >
> > > > Despite the fact that agents only see the sum of the losses of these two games, D3C was able to tease out the correct teams (players 1 and 2 vs 3 and 4). By the way, we fixed the makeshift diagram for this game in our previous response. We did not realize the first time that the formatting did not work as planned.
> > > >
> > > > 2.
> > > > Indeed, given your comment we agree that we have extended the "incentive compatible" term beyond its standard meaning in the mechanism design literature. Agents could manipulate by sharing or behaving according to false loss functions. Our result is that if all agents opt into the mechanism (with their true loss functions), they achieve good outcomes. And if some agents opt into the mechanism, in some settings (e.g., see Appendix E.2.2), the best thing an agent can do is also opt in. In other words, opting into the mechanism and acting according to the true loss function and the proposed system is a Nash equilibrium.
> > > >
> > > > Regarding the specific conditions for agents to reach a Nash, in Section 2.7 “Assessment” we state “ If each agent’s original loss is convex with diagonally dominant Hessian and the strategy space is unconstrained, the unique, globally stable fixed point of the game defined with mixed losses is a Nash (see Appendix H.4).” Note this is just to say that the resulting joint strategy $x^*$ is a Nash equilibrium given a fixed $A$, not that the pair $(x^*, A^*)$ is a Nash.

---

### Official Review · AnonReviewer4 · 2020-10-28
**recommendation to accept**

**Rating:** 6
**Confidence:** 3

**Review:**

##########################################################################

Summary:

This paper proposes a differentiable, local estimator of multi game inefficiency, as measured
by price of anarchy. We then present two instantiations of a single decentralized meta-algorithm, one
1st order (gradient-feedback) and one 0th order (bandit-feedback), that reduce this inefficiency. Experiments conducted in three domains (traffic network, Coins and Cleanup) show that agents minimizing local estimates of price of anarchy achieve lower loss on
average than selfish agents without the ability to compromise.

##########################################################################

Reasons for score:

Overall the paper is well organized and provided clear motivation of the problem. The proposed approach is well-presented and the idea of mixing losses is interesting. An upper bound for the optimized price of anarchy is presented.

##########################################################################Pros:

Detailed comments and concerns:

1. What is the implication of "mixing losses" in practical systems? What are the incentives for agents to adjust their original incentives? I would like to see more discussions addressing the incentive compatibility issue in adjusting for a mixing losses.

2. The authors show that if each agent’s original loss is convex with diagonally dominant Hessian and the strategy space is unconstrained, the the unique, globally stable fixed point of the game defined with mixed losses is a Nash. I would be willing to see more discussions on if these restrictions could be relaxed to achieve Nash under the mixed losses.

3. What are the computation times for running D3C, and how does that compare to the baselines?

----- Post Discussion ---- Updates to the paper have helped make the motivation/technical parts of the paper more clear. The authors' response are helpful w.r.t the questions I raised in the review.

---

> ### Author Response · Authors · 2020-11-16
> **Response to Reviewer 4**
>
> Thank you for reviewing our paper. We’re glad to hear that you found it well motivated and presented. We’ll respond to your comments bullet by bullet.
>
> 1.
> a) Practically speaking, “mixing losses” means that after every episode (or policy learning update loop) each agent must broadcast their return (discounted sum of rewards, i.e., a single float), $G_i$, to a central master that then mixes them (i.e., performs the matrix vector product $A^T G$ where $A$ is the mixing matrix and $G = [G_1, \ldots, G_N]^\top$). Every few episodes, an agent updates their mixing weights and sends them to the master to overwrite the row of $A$ corresponding to that agent. Note that instead of broadcasting to a central master, the agents could broadcast the necessary weights directly to each other and mixing could be performed by each individual agent -- this is technically equivalent but maybe a philosophically interesting distinction.
> b) Please take a second look at the introduction where we discuss how "We appeal to individual rationality in three ways." Also look at Section 2.7 starting with "We justify the agents learning weight vectors..." which explains how the proposed meta-algorithm addresses these concerns. Please let us know any specific questions you have after reviewing these.
>
> 2.
> We don't currently see any route to relaxing this restriction. The fact that the mixture weights and strategies are adjusted concurrently results in nonlinear dynamics and proving these continuous time dynamics (let alone discrete updates) converge to a Nash is very difficult. It may be possible to prove local convergence as is done with GANs where the game is also highly nonlinear.
>
> 3.
> Maybe the answer to this question is more clear after reading our answer to 1a. The computation required for mixing losses is negligible in our experiments (computing a single matrix-vector product $A^T$ [vector_of_losses]) and so the runtime is the same. If we were to scale up to 1000s of agents, runtime may be affected, but would likely still be dominated by the environment step and querying the neural network policy. Note that our experiments plot the losses and returns of the agent after every strategy (policy) update, not every mixture weight update so the comparison to a vanilla selfish agent really is “apples to apples”. In other words, if a loss-sharing agent tries out a mixing weight for 5 steps during which it makes 5 policy updates, we plot the returns for each policy update. A vanilla selfish agent does the same; it just does not ever adjust its mixing weight (it is fixed to a onehot).

---

### Official Review · AnonReviewer2 · 2020-10-30

**Rating:** 7
**Confidence:** 3

**Review:**

This paper proposes a (decentralized) method for online adjustment of agent incentives in multi-agent learning scenarios, as a means to obtain higher outcomes for each agent and for the group as a whole. The paper uses the “price of anarchy” (the worst value of an equilibrium divided by the best value in the game) as a proxy for the efficiency of the game outcome, and derive an upper bound on a local price of anarchy that agents can differentiate. In several experiments (a traffic network, the coin game, Cleanup), their method leads to improved individual agent and group outcomes relative to baselines, while avoiding cases of stark division of labor that sometimes emerges when agents directly optimize the sum of all agent rewards.

Pros:
Overall, I thought this paper was quite strong. I agree with the claim in the paper that having agents simply optimize the cooperative return is not always realistic or interesting. The framing of compromise as the mixing of agent incentives is particularly interesting, and it makes intuitive sense to me. I’m not familiar enough with the literature to know if this formulation is novel (and unfortunately the paper does not have a Related Work section). The paper makes significant contributions towards making this idea practical, including relaxing the requirement that the agents can observe or directly differentiate with respect to the other agent strategies.

The paper is also pretty well written and communicated, though I did not dive into the proofs and skimmed some of the technical details. I appreciated the frankness with which the paper describes their method, e.g. in the following excerpts:
“Ideally, one meta-algorithm would allow a multi-agent system to perform sufficiently well in all these scenarios. The approach we propose, D3C (Sec. 2), is not that meta-algorithm, but it represents a holistic effort to combine critical ingredients that we hope takes a step in the right direction”
“We also provide specific, albeit narrow, conditions under which agents may achieve a Nash equilibrium”

Cons:
Of course, the ‘mixing of losses’ strategy proposed in the paper requires that agents be able to observe the losses of other agents, which is not always feasible in practice.

Other notes:
I was a bit confused by the sentence: “Our agents reconstruct their losses using the losses of all other agents as a basis”, which seems to imply that the agents’ own loss function is not used as part of this basis.

The Appendix also seems to be missing from the paper, although there are references to it in the text (e.g. reference to F.4).

Overall:
On the whole, I think this paper is quite good and worthy of acceptance.

---

> ### Author Response · Authors · 2020-11-16
> **Response to Reviewer 2**
>
> Thank you for taking the time to carefully read our paper. We’re pleased to hear that the major message resonates with you.
>
> Pros:
>
> We have a longer version of the paper prepared with a distinct related work section that touches on connections to recent work on learning loss functions, learning loss mixtures, gifting rewards to other agents, and more general connections to social psychology, neuroscience, and evolutionary biology, so we will include this in our revision. FYI, there is another submission to ICLR titled “Learning to Share in Multi-agent Reinforcement Learning” which also looks at mixing losses, but it is motivated from Markov games rather than from a classical game theoretic perspective like ours (price of anarchy, Nash, Myerson-Satterthwaite theorem). We have no relation to that paper; we just noticed it and thought you might find it interesting. Their related work section highlights the lack of decentralized approaches and so has a similar motivation to our work.
>   Our work is most similar to “Inducing Cooperation through Reward Reshaping based on Peer Evaluations in Deep Multi-Agent Reinforcement Learning“ [AAMAS 2020]. This work also provides a decentralized approach that transforms the game by modifying rewards, however, it does not guarantee “budget-balance” nor does it derive its proposed algorithm from any principle (e.g., price of anarchy); the proposed algorithm is a heuristic supported by experiments. Other work, “Gifting in Multi-Agent Reinforcement Learning“ [AAMAS 2020] explores gifting rewards to agents as well, but it does so by simply expanding the action space of agents to include a gifting action. It is also not budget balanced.
>
> Cons:
>
> As you say, our proposed approach is not ideal for all settings. There are cases where sharing reward scalars might be prohibitively expensive (or not permitted) and agents may have to learn to cooperate and circumvent social dilemmas using only access to their own environment observations. We will also compare this assumption to others made in the literature (observing other agent actions or access to other agent policies is common, e.g., “Learning to Incentivize Others (LIO)” / ”Learning with Opponent-Learning Awareness (LOLA)”). Note that among the possible bits of information to communicate-- policy weights ($w_i \in \mathbb{R}^d$), observations ($o_i \in \mathbb{R}^m$), actions/strategies ($x_i \in \mathbb{R}^{q}$ for continuous actions), rewards ($r_i \in \mathbb{R}$)-- we think rewards are possibly the cheapest. Note, communicating actions is cheap for discrete action spaces, but expensive for continuous (vector) spaces. On the other hand, rewards are always scalars. It’s amazing how much a human can communicate with a frown (-1) or smile (+1) ;)
>
> Other notes:
>
> Thanks for raising this. We should have said "all agents" instead of "all other agents". That was a mistake.
>
> The appendix can be found in the supplementary material. There should be a “download arrow” next to the word “zip” at the top of the OpenReview page.

---

### Author Response · Authors · 2020-11-23
**Changes to the Paper**

We have made a few changes to the paper in line with some of our responses below. Specifically, we

- Fixed the typos noticed by the reviewers and made a few minor edits to the writing,
- Added the explicit dependence of $x$ on $t$, i.e., $x=x(t)$, to the notation in Section 2.1,
- Fixed / removed the statement regarding PPAD-complete below Equation 1,
- Added a footnote regarding the dependence of $\Delta t$ on $\beta$ in Theorem 1,
- Added additional "Relative Reward Attention" plots to Figure 6,
- Included a separate related work section in Section 4.

We are very grateful for the reviewers' engagement in the discussion period. This was extremely helpful for improving the paper and we appreciate the time you took to thoughtfully consider and respond to our rebuttals.

---

### Decision · Program_Chairs · 2021-01-07
**Final Decision**

**Decision:**

Reject

**Comment:**

This paper proposes a method of decentralized mechanism design to reduce the price of anarchy. Based on the detailed responses of the authors, all reviewers were satisfied by the technical contribution after the rebuttal period.

There was, however, a heavily engaged and lengthy discussion between most reviewers regarding the applicability of the method and how it links to the motivation given in the paper. The paper could be improved by (1) highlighting an exemplar real world use case in the paper motivation (there are a couple mentioned briefly in the introduction but one of these could be emphasized more); and (2) connecting the choices made in the design of the approach to the opening motivation sections and exemplar use case.

The level of engagement from most reviewers demonstrates a good level of interest from a representative sample of the ICLR community but demonstrated that their remained work outstanding to clarify the core message and significance of the contribution.